# Human flourishing and religious liberty: Evidence from over 150 countries

**Christos Andreas Makridis** *

MIT Sloan School of Management, Cambridge, Massachusetts, United States of America

* makridis@mit.edu

## Abstract

This paper studies the spatial and time series patterns of religious liberty across countries and estimates its effect on measures of human flourishing. First, while there are significant cross-country differences in religious liberty, it has declined in the past decade across countries, particularly among countries that rank higher in economic freedom. Second, countries with greater religious liberty nonetheless exhibit greater levels of economic freedom, particularly property rights. Third, using micro-data across over 150 countries in the world between 2006 and 2018, increases in religious freedom are associated with robust increases in measures of human flourishing even after controlling for time-invariant characteristics across space and time and a wide array of time-varying country-specific factors, such as economic activity and institutional quality. Fourth, these improvements in well-being are primarily driven by improvements in civil liberties, such as women empowerment and freedom of expression.

## Introduction

THE UNITED STATES WAS FOUNDED by individuals fleeing religious persecution from the Anglican church in Great Britain. These individuals who initially settled the country, and eventually the Founding Fathers, not only did not distinguish between civil and religious liberty, but also viewed religious liberty as humanity's most fundamental form of freedom [1].

While there is a large literature on the effects of regulation and property rights on economic and social development, there is a much smaller literature on the role of religious liberty. This comes at a time when nearly 80% of people throughout the world live in a "religiously restricted environment," prompting the Department of State, and the United States at large, to champion religious liberty as a national and economic security priority [2].

Descriptive evidence suggests that countries with greater religious liberty have greater levels of economic development [3, 4]. However, whether such a relationship is causal is a challenging question because of two confounding forces that move in opposite directions. On one hand, countries with greater religious liberty may also have better economic institutions, like property rights, that promote economic development and human flourishing [5–7]. On the other hand, religious affiliation is negatively associated with economic growth [8, 9]. The primary contribution of this paper is to explore whether such a plausibly causal relationship

**Data Availability Statement:** The data underlying the results presented in the study are available from Gallup (contact Kris Hodgins at Kris_Hodgins@gallup.com) for researchers who meet the criteria for access to confidential data. Interested researchers can replicate the study findings in their entirety by directly obtaining the

data from Gallup Organization and following the protocol in the Methods section. While author CAM's role as a senior adviser at Gallup allows for access to the World Poll for free, other researchers can obtain a license to work with the data. Moreover, researchers at a handful of universities may already have free access to the data.

**Funding:** The authors received no specific funding for this work.

**Competing interests:** The authors have declared that no competing interests exist.

between religious liberty and well-being exists. Fig 1 provides suggestive evidence that the answer is yes: the countries that experienced the greatest growth in religious liberty between 2006 and 2018 also experienced the greatest growth in human flourishing, which we define and investigate in the paper ahead.

The first part of the paper documents two stylized facts about religious liberty and its correlation with various measures of economic development using the Gallup World Poll, the Varieties of Democracy ("V-Dem"), and the World Bank. First, the average (median) country has experienced an 8% (13%) decline in religious liberty between 2006 and 2018. These declines are, perhaps counterintuitively, concentrated among countries with greater economic freedom, especially those with stronger property rights. Second, a strong positive correlation between religious liberty and economic freedom nonetheless exists in the cross-section, which is related with a large literature on the role of institutions for economic development [7, 10]. These results are also consistent with prior literature that religious freedom is positively associated with almost all of the pillars of global competitiveness in the World Economic Forum's Global Competitiveness Index [3, 4], suggesting that religious liberty is a prerequisite, or at least a complementary factor, for other forms of economic development and economic freedom.

The second part of the paper quantifies a plausibly causal effect of religious liberty on human flourishing. Using year-to-year variation in changes in the social and governmental regulation of religion between 2006 and 2018, a standard deviation (sd) increase in religious liberty is associated with a 0.03 percentage point (pp) rise in the probability that an individual is thriving and a 0.08sd rise in individual well-being. These improvements in human flourishing are concentrated among religious minorities. To put these estimates in perspective, since only 26% of the sample reports that they are thriving, the marginal effect of a standard deviation change in religious liberty—equivalent to transforming a country like Russia into the United States—amounts to approximately an 11% increase in the share of thriving individuals.

The baseline specification controls for not only country and year fixed effects, but also a wide array of demographic and country-specific time-varying factors, such as economic growth and institutional quality. While these controls help mitigate the concern that there are unobserved shocks affecting human flourishing that are also correlated with religious liberty, they may fail to control for the potential negative association between economic growth and religious affiliation that has been suggested in prior literature. As an alternative identification strategy, I exploit plausibly exogenous historical variation in the exposure of missionaries across countries [11]. The identifying assumption is that exposure to missionaries prior to 1923 increases the probability that a country has more egalitarian norms governing religious liberty today, but does not affect human flourishing through other channels besides religious liberty. The resulting estimates are larger than the least squares estimates, suggesting that the baseline provides a lower bound.

These results could be consistent with at least one of three different mechanisms. First, religious liberty could lead to an increase in well-being through its effects on democratic governance and freedom of expression since individuals are empowered to believe what they want and participate in the civics process. Second, by placing an emphasis on religious pluralism and competition among different worldview for the pursuit of truth, it could lead to greater educational attainment. Third, it could reduce the probability of entering into civil war and other forms of armed conflict. Using additional data on these country-specific outcomes between 1995 and 2018, the results are primarily consistent with the first mechanism: increases in religious liberty affect well-being through its effects on democratic governance and freedom. For example, there are economically and statistically strong positive associations between religious liberty and civil liberties, women empowerment, access to justice, and freedom of

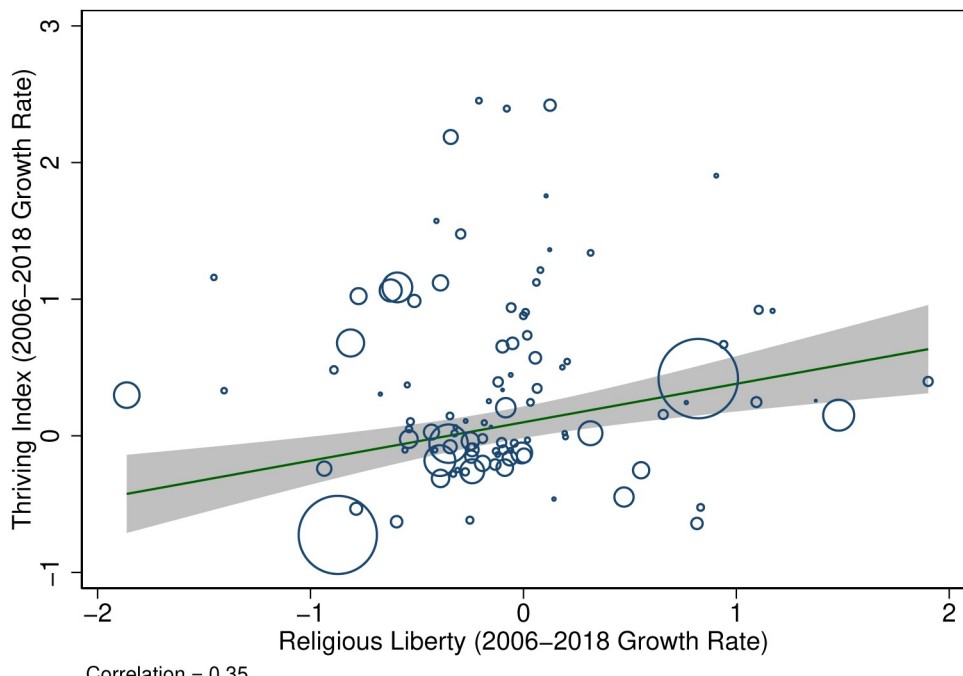

**Fig 1. Motivating evidence on religious liberty and human flourishing.** Sources: Gallup World Poll, Varieties of Democracy, 2006-2018. The figure reports the population-weighted association between the 2006-2018 growth rate in religious liberty and the share of individuals who report that they are thriving. This measure of thriving is generated based on having a response of 7 or higher on a 10-point scale about current life satisfaction and having a response of 8 or higher on a similar 10-point scale about expected future life satisfaction in 5 years (see Table 1).

expression, as well as some negative associations with public and political corruption even after controlling for time-varying measures of economic performance and both country and year fixed effects. These results are consistent with contributions on the complementarity of social capital and democratic institutions [12–14].

This paper contributes to two literatures. The first investigates the role of institutions on economic development [15]. Although an active debate remains over the exact quantitative effects of institutions on economic development, there is causal evidence documenting the importance of property rights [7], regulation [16], and the adverse consequences of colonialism [6, 10] for economic development. There is also a large political science literature about the effects of democratic institutions on stability [17] and economic growth [18].

However, beyond these empirical investigations of political and economic institutions, there is a much smaller literature about the effects of religious institutions on measures of human flourishing and economic development [19]. This paper contributes to an ongoing debate about the effects of religious pluralism on religious vitality. Some evidence supports "the supply-side view"—that greater religious pluralism was associated with greater church attendance in the United States [20–22] and across the world [23–26]—a vigorous debate remains over whether the relationship is negative [27–29] or potentially null [30]. By finding that increases in religious liberty lead to increases in human flourishing, this paper provides new microeconomic evidence consistent with the supply-side view that religious participation and well-being thrive under pluralism.

The second literature investigates the effect of culture (e.g., religious affiliation) and institutions on economic development [31]. There has been a general recognition that religious practices and beliefs affect economic activity at a microeconomic level [32], ranging from

subjective well being [33] to criminal activity [34] and educational attainment [35]. These theories also have aggregate implications, including whether religious affiliation is associated with economic growth on a panel of countries [8]. Similarly, motivated by evidence that more religious people tend to be more trusting and more trustworthy [36], some have found that religious affiliation is negatively associated with attitudes about innovation [37]. However, causal inference has been challenging [38].

Using plausibly exogenous fluctuations in the lunar cycle and the resulting hours in a day, one study found a negative association between fasting during Ramadan and economic growth [9]. While the identification strategy allows for a more causal interpretation than traditional estimates, one limitation is that the identifying variation comes from only small movements in the hours available each day, making it challenging to extrapolate out of sample. Moreover, their analysis is restricted to the set of Muslim adherents across countries. This paper contributes to the ongoing debate about the economic effects of religious affiliation by focusing more broadly on religious liberty, which is important for protecting against religious persecution [39]. Moreover, this paper draws on much more comprehensive data that covers over 150 countries annually for over a decade, contrasting with prior studies that have used the World Values Survey (WVS), which covers roughly 50 countries with smaller samples and fewer questions that reflect the concept of human flourishing [40].

The structure of the paper is as follows. Section 2 introduces the new data and measurement strategy. Section 3 describes the empirical strategy and presents the main results. Section 4 investigates the mechanisms behind this result. Section 5 concludes.

## Data, measurement, and key patterns

### Repeated cross-section of well-being (2006-2018)

The primary dataset in this analysis is the World Gallup Poll, which contains surveys from over 150 countries that make up 98% of the world's population based on randomly selected and nationally representative samples. While these surveys are launched multiple times a year in most countries, all countries have the survey administered at least once, barring severe extenuating circumstances. The baseline empirical specification pools all countries together, but the results are robust to restricting the sample to countries observed at least 11 times, as well as to a fully balanced panel, although the standard errors rise marginally. (The matched data includes responses from 156 countries observed for at least five years between 2006 and 2018 with between 290 and 11,420 respondents within a given year depending on the country. Roughly 80% of countries are observed at least 11 times and 47% are observed all 13 years.) Survey questions are designed to cover a wide array of key indicators, including law & order, food & shelter, job creation, migration, financial well-being, personal health, civic engagement, and evaluative well-being.

Each questionnaire is translated into the major conversational language in each country. To maximize accessibility, two approaches can be used. The first approach involves completing two independent translations with an independent third party who also has some knowledge of survey research methods who adjudicates the differences. A professional translator will subsequently translate the final version back into the source language. The second approach involves using a translator to translate the survey into the target language and an independent translator back into the source language. An independent third party with knowledge of survey methods will review and make any final translation modifications. Interviewers for each country are instructed to follow the script and not to deviate from the translated language.

Gallup selects quality vendors with experience in survey design and implementation with in-depth training sessions with local field staff prior to the start of data collection. Gallup also

follows ESOMAR standards for quality control. A supervisor accompanies each interviewer for one full interview within the first two days of interviewing and the supervisor accompanies interviews on a minimum of 5% of subsequent interviews. Interviewers re-contact a minimum of 15% of households to ensure correct execution of random route procedures and within-household selection. Telephone surveys are used in countries where coverage represents at least 80% of the population. Information that is gathered is also standardized so that it is comparable across countries, e.g., education (elementary, secondary, and tertiary) and income.

Table 1 enumerates the questions used in the survey design to measure well-being. Two main measures are employed. The first is an indicator for whether an individual reports that they are thriving. As Table 1 describes, individuals are surveyed on a scale of 0 to 10 about their current and expected future (in five years) life satisfaction. If an individual reports at least a 7/10 on current life satisfaction and at least an 8/10 on expected future life satisfaction, they are classified as thriving. The second conducts a principal component analysis (PCA) over four standard normal measures of subjective well-being: daily experience, optimism, positive experience, and negative experience. The resulting latent index extracts the common signal among each of these four measures, which is more likely to reflect the broader concept of human flourishing [40]. Nonetheless, the results are statistically indistinguishable from those obtained from a simpler heuristic, like the unweighted average across each of the four sub-indices, is used.

Both these measures are comparable with the measures of happiness from the World Values Survey. For example, some use an indicator for whether an individual reports "quite happy" or "very happy" in response to the question: "Taking all things together, would you say you are: not at all happy, not very happy, quite happy, very happy?" [9]. Similarly, some also create an indicator for whether the respondent reports a value above 5 in response to the question "How satisfied are you with your life as a whole these days," which is asked on a numerical 10-point scale. In this sense, while the Gallup data provides several comparable measures to the WVS, it also provides a wide range of additional responses about institutions, human flourishing, and religious affiliation.

## Time-varying measure of religious liberty and institutions

While there are several sources of information on religious liberty available, like the Pew Research Center's measurement of government restrictions and social hostilities [41], the Varieties of Democracy ("V-Dem") has emerged as the first attempt to measure a wide array of cross-country institutional characteristics on a consistent and continuous basis year-after-year using a combination of state-of-the-art statistical methods and expert elicitation [42].

Unlike standard approaches that produce an index based on the response to specific questions, V-Dem treats each measure as an inherently latent variable that (approximately five) expert raters only observe manifestations of in a noise environment. This approach, known as differential item functioning (DIF), perceive latent regime characteristics that map into ordinal scales in V-Dem. Recognizing that each expert will interpret questions differently, V-Dem allows for the possibility that raters apply different thresholds when they map their perceptions of latent traits into ordinal ratings for each of the different measures. As long as the errors in each of these measurements are uncorrelated with each other, the covariance across them will identify the latent distribution.

One concern with this approach is that experts may vary in non-random ways across countries. For example, if countries with lower productivity attract lower quality experts, then the ratings for some countries might be lower than others and correlate in unobserved ways with human flourishing. However, because V-Dem allows raters to apply different thresholds when

**Table 1. Definitions of human flourishing measures.**

| Variable index | Definition |
|---|---|
| *Thriving* | The Thriving Index measures respondents perceptions of where they stand on a ladder scale with steps numbered from 0 to 10, where "0" represents the worst possible life and "10" represents the best possible life. Individuals are "thriving" if they say they presently stand on step 7 or higher of the ladder and expect to stand on step 8 or higher five years from now. Please imagine a ladder, with steps numbered from 0 at the bottom to 10 at the top. The top of the ladder represents the best possible life for you and the bottom of the ladder represents the worst possible life for you. On which step of the ladder would you say you personally feel you stand at this time? Individuals who rate their current lives a "7" or higher AND their future an "8" or higher are "thriving." Individuals are "suffering" if they report their current AND future lives as a "4" and lower. |
| *Daily Experience* | The Daily Experience Index is a measure of respondents' experienced well-being on the day before the survey. The index provides a real-time, composite measure of respondents' positive and negative experiences. (i) Did you feel well-rested yesterday? (ii) Were you treated with respect all day yesterday? (iii) Did you smile or laugh a lot yesterday? (iv) Did you learn or do something interesting yesterday? (v) Did you experience the following feelings during a lot of the day yesterday? How about enjoyment? (vi) Did you experience the following feelings during a lot of the day yesterday? How about physical pain? (vii) Did you experience the following feelings during a lot of the day yesterday? How about worry? (viii) Did you experience the following feelings during a lot of the day yesterday? How about sadness? (ix) Did you experience the following feelings during a lot of the day yesterday? How about stress? (xi) Did you experience the following feelings during a lot of the day yesterday? How about anger? |
| *Optimism* | The Optimism Index measures a respondent's positive attitude for the future. Specifically, respondents are asked whether certain aspects of their life are getting better or getting worse. (i) Right now, do you feel your standard of living is getting better or getting worse? (ii) Right now, do you think that economic conditions in the city or area where you live, as a whole, are getting better or getting worse? (iii) Please imagine a ladder, with steps numbered from 0 at the bottom to 10 at the top. The top of the ladder represents the best possible life for you and the bottom of the ladder represents the worst possible life for you. Just your best guess, on which step do you think you will stand in the future, say about five years from now? |
| *Positive Experience* | The Positive Experience Index is a measure of experienced well-being on the day before the survey. Questions provide a real-time measure of respondents' positive experiences. (i) Did you feel well-rested yesterday? (ii) Were you treated with respect all day yesterday? (iii) Did you smile or laugh a lot yesterday? (iv) Did you learn or do something interesting yesterday? (v) Did you experience the following feelings during a lot of the day yesterday? How about enjoyment? |
| *Negative Experience* | The Negative Experience Index is a measure of experienced well-being on the day before the survey. Questions provide a real-time measure of respondents' negative experiences. (i) Did you experience the following feelings during a lot of the day yesterday? How about physical pain? (ii) Did you experience the following feelings during a lot of the day yesterday? How about worry? (iii) Did you experience the following feelings during a lot of the day yesterday? How about sadness? (iv) Did you experience the following feelings during a lot of the day yesterday? How about stress? (v) Did you experience the following feelings during a lot of the day yesterday? How about anger? |

Sources: Gallup World Poll, 2006-2018. The table documents the survey questions used for measuring human flourishing as done by the empirical strategy. Index scores are calculated among individuals. For each individual, the items are recoded so that positive answers are scored as a "1" and all other answers (including don't know and refused) are scored as a "0." If a record has no answer for an item, then that item is not eligible for inclusion in the calculations. An individual record has an index calculated if it has at least four out of five valid scores (0 or 1). Or, in the case where there are 10 items, at least 8 must be answered (daily experience); in the case where there are 3 items, 2 must be answered (optimism); in the case where there are 5 items, 4 must be answered (positive and negative experience).

they map their perceptions into the ordinal scales, V-Dem produces a standard deviation of each measurement, which can serve as a control for potential classical or non-classical measurement error in the index of interest.

This paper focuses on the measurement of one specific index from V-Dem: religious liberty. The survey question on religious liberty asks about "the extent to which individuals and groups

**Table 2. Measuring religious liberty across countries.**

| Rating | Response |
|--------|----------|
| 0 | Not respected by public authorities. Hardly any freedom of religion exists. Any kind of religious practice is outlawed or at least controlled by the government to the extent that religious leaders are appointed by and subjected to public authorities, who control the activities of religious communities in some detail. |
| 1 | Weakly respected by public authorities. Some elements of autonomous organized religious practices exist and are officially recognized. But significant religious communities are repressed, prohibited, or systematically disabled, voluntary conversions are restricted, and instances of discrimination or intimidation of individuals or groups due to their religion are common. |
| 2 | Somewhat respected by public authorities. Autonomous organized religious practices exist and are officially recognized. Yet, minor religious communities are repressed, prohibited, or systematically disabled, and/or instances of discrimination or intimidation of individuals or groups due to their religion occur occasionally. |
| 3 | Mostly respected by public authorities. There are minor restrictions on the freedom of religion, predominantly limited to a few isolated cases. Minority religions face denial of registration, hindrance of foreign missionaries from entering the country, restrictions against proselytizing, or hindrance to access to or construction of places of worship. |
| 4 | Fully respected by public authorities. The population enjoys the right to practice any religious belief they choose. Religious groups may organize, select, and train personnel; solicit and receive contributions; publish; and engage in consultations without undue interference. If religious communities have to register, public authorities do not abuse the process to discriminate against a religion and do not constrain the right to worship before registration. |

Sources: Varieties of Democracy. The table reports the potential responses from expert solicitations in response to the question "is there freedom of religion?"

have the right to choose a religion, change their religion, and practice that religion in private or in public as well as to proselytize peacefully without being subject to restrictions by public authorities." Table 2 documents the ordinal scale that range between zero and four, together with the corresponding responses from the experts who are being surveyed.

This approach is a form of item response theory (IRT), which provides ways for dealing with disagreement among experts over inherently subjective assessments. Although they are much more complex than simple heuristics, like an average across survey respondents, they exhibit much better performance [43]. One reason is the ability to control and explicitly model survey respondent reliability. Because experts in the V-Dem survey answer multiple questions, multiple answers that deviate from the norm will produce a lower reliability, thereby generating lower weights in the inclusion of the respondent's answer in the overall score.

To understand how these data compare with more conventional sources that are only available in the cross-section (i.e., not panel), Fig 2 plots the V-Dem measure of religious liberty with the two indices of government restrictions and social hostilities from the Pew Research Center. These variables are presented in their standard normal form with a mean of zero and standard deviation of one. Not surprisingly, there are strong negative correlations between the two, particularly between government restrictions and V-Dem (with a correlation of -0.77). The fact that the V-Dem data is negatively correlated with both of these indices suggests that it is capturing features of both formal government policies that restrict religion and informal social hostilities that create pressure against religious freedom and pluralism.

## World bank panel of country characteristics

The main supplementary data is from the World Bank, which contains time-varying country economic and demographic characteristics, such as GDP and employment growth, and the Heritage Foundation Index of Economic Freedom, which contains time-varying country

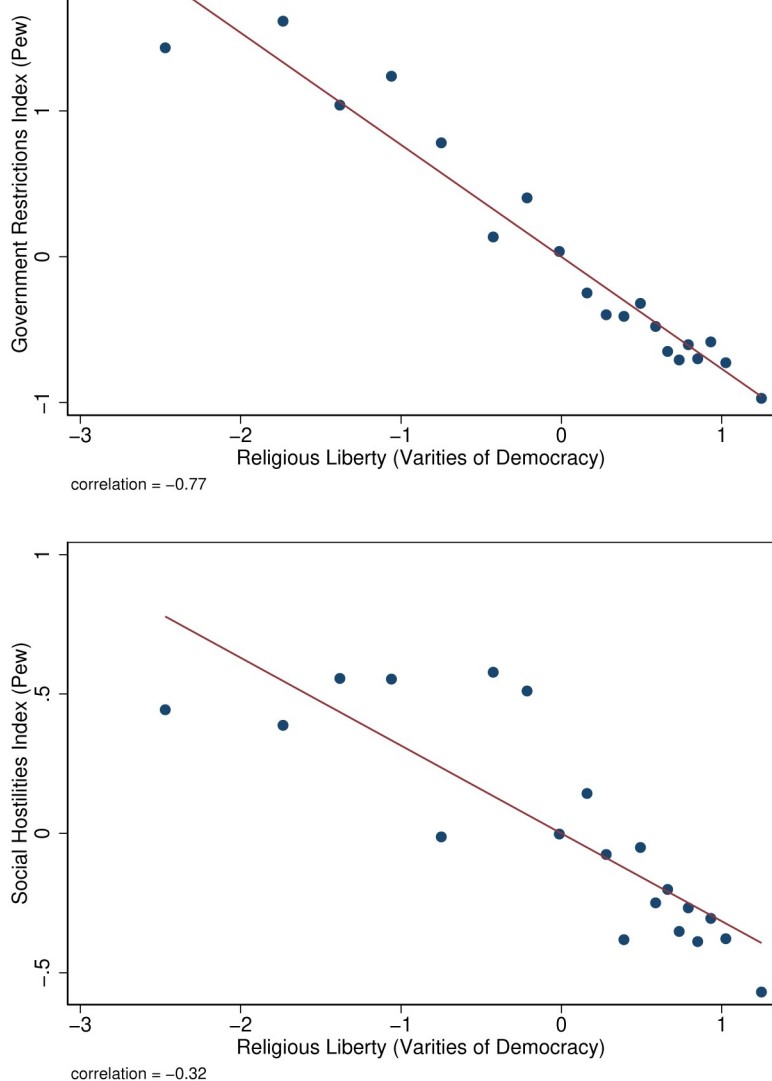

**Fig 2. Comparison of varieties of democracy and Pew research data.** Sources: Varieties of Democracy ("V-Dem")
Pew Research Center, 2007, 2016, 2017. The figures plots the standardized z-scores of religious liberty from V-DEM
and government restrictions and social hostilities from the Pew Research Center.

measures of institutional quality. Comparable with the Fraser Institute, these variables are
important for not only controlling covariates in the main regression, but also determining
mediating influences [44]. For example, changes in economic growth are often precipitated by
changes in economic freedom [45].

Does religious liberty simply proxy for other, more traditional measures of institutional
quality? While the next section will explore some bivariate correlations, Table 3 begins by esti-
mating cross-sectional regressions of religious liberty on various dimensions of economic free-
dom (as $z$-scores), controlling for a few time-varying country characteristics, such as GDP
growth and population. Here, overall economic freedom consists of four categories: rule of law
(property rights, government integrity, judicial effectiveness), government size (government
spending, tax burden, fiscal health), regulatory efficiency (business freedom, labor freedom,

**Table 3. Religious liberty and economic freedom, 1996-2018.**

| | Religious liberty (z-score) | | | | | |
|---|---|---|---|---|---|---|
| | (1) | (2) | (3) | (4) | (5) | (6) |
| Economic freedom (z-score) | .24*** | | | | | .06 |
| | [.06] | | | | | [.14] |
| Property rights (z-score) | | .25*** | | | | .48** |
| | | [.07] | | | | [.19] |
| Government integrity (z-score) | | | .19*** | | | -.18 |
| | | | [.06] | | | [.12] |
| Judicial effectiveness (z-score) | | | | .18** | | -.07 |
| | | | | [.09] | | [.13] |
| Business freedom (z-score) | | | | | .13** | -.08 |
| | | | | | [.06] | [.12] |
| GDP growth | -1.19** | -.84 | -.89 | -.21 | -.89 | -1.20 |
| | [.51] | [.53] | [.54] | [3.38] | [.54] | [2.72] |
| log(Population) | .00 | -.00 | -.00 | -.05 | -.00 | -.03 |
| | [.04] | [.04] | [.04] | [.05] | [.04] | [.05] |
| Agricultural share | 1.75** | 1.85** | 1.79** | 2.78*** | 1.86** | 3.34*** |
| | [.85] | [.85] | [.88] | [1.03] | [.86] | [1.01] |
| Services share | 2.11* | 2.13* | 2.15* | 3.71*** | 2.50** | 4.26*** |
| | [1.20] | [1.20] | [1.26] | [1.41] | [1.20] | [1.32] |
| R-squared | .32 | .32 | .30 | .25 | .29 | .31 |
| Sample Size | 3539 | 3550 | 3562 | 317 | 3561 | 311 |
| Country Controls | Yes | Yes | Yes | Yes | Yes | Yes |

Sources: Heritage Foundation, Varieties of Democracy, 1996-2018. The table reports the coefficients associated with regressions of standardized religious liberty on measures of standardized institutional quality, controlling for time-varying country measures of demographics and economic productivity. Standard errors are clustered at the country-level and observations are unweighted.

monetary freedom), and open markets (trade freedom, investment freedom, and financial freedom).

There is a strong positive association for each of these measures: a 1sd rise in overall economic freedom is associated with a 0.24sd rise in religious liberty, conditional on controls. Interestingly, property rights are most closely correlated with religious liberty. In fact, it is the only characteristic of economic freedom that systematically enters in a statistically and economically significant way across specifications, particularly in column 6 where each dimension is included as a control. These results suggest that, while economic freedom is correlated with religious liberty, the latter is detecting a fundamentally different dimension of institutional quality, consistent with existing cross-sectional evidence [4].

While religious liberty is correlated with other measures of institutional quality, it is capturing systematically different features across countries. Using an average between 1995 and 2018, Fig 3 plots population-weighted correlations between standardized religious liberty and four measures of institutions: property rights, government integrity, business freedom, and labor freedom. The strongest correlation is between religious liberty and property rights, which is 0.67 (Panel A), suggesting that areas with stronger ownership over property rights also exhibit greater religious liberty. The correlations with government integrity and business freedom are 0.43 and 0.51 (Panels B and C), respectively. (The corresponding population-weighted correlation with the logarithm of per capita GDP is 0.30.) These results are consistent with prior literature that religious freedom is positively associated with almost all

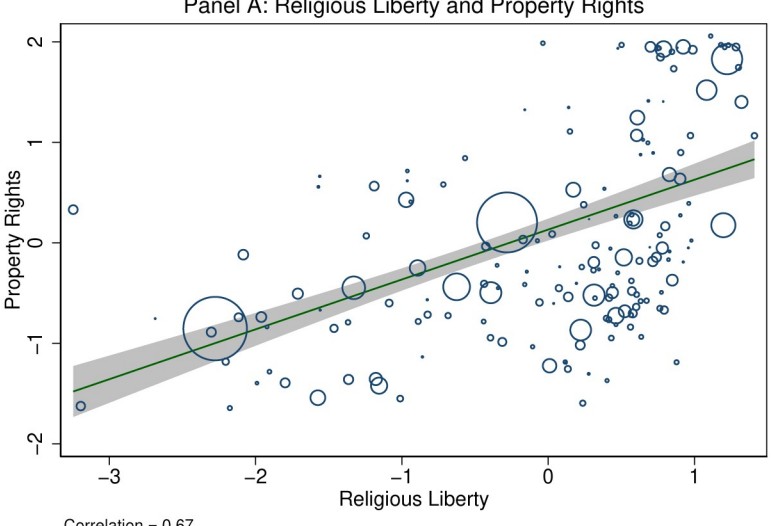

Panel A: Religious Liberty and Property Rights

Correlation = 0.67

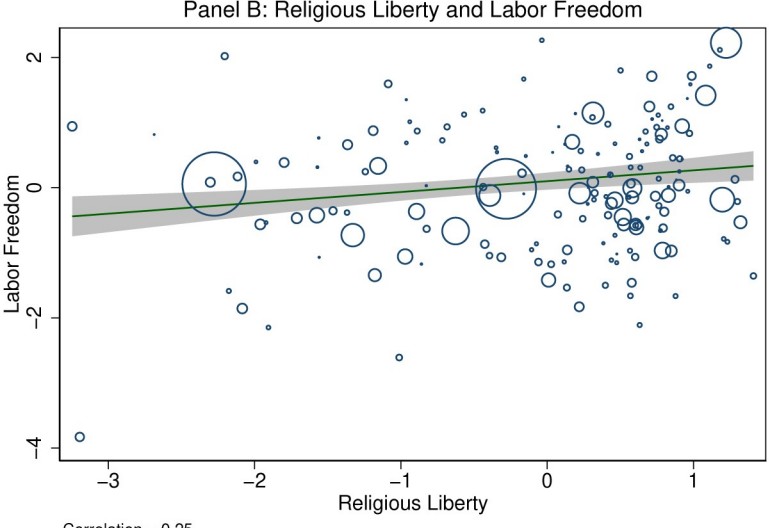

Panel B: Religious Liberty and Labor Freedom

Correlation = 0.25

**Fig 3. Religious liberty and heritage index of economic freedom correlations.** Sources: Varieties of Democracy and Heritage Foundation. The figure plots the population-weighted relationship between standard normal measures of religious liberty and institutional quality measured with property rights and labor market freedom. These indices are standardized based on their 1995-2018 average.

of the pillars of global competitiveness in the World Economic Forum's Global Competitiveness Index [3, 4].

Moreover, the V-Dem data contains significant within-country variation. To provide a characterization of this variation, Fig 4 plots the distribution of growth rates in religious liberty indices between 2006 and 2018 across countries. However, there has been a large decline in religious liberty over these years—a mean of 8.1% and a median of 13.25%. Importantly, however, the fact that there is such wide variation (standard deviation = 0.57) is important for the identification strategy, which exploits year-to-year changes in religious liberty and human flourishing.

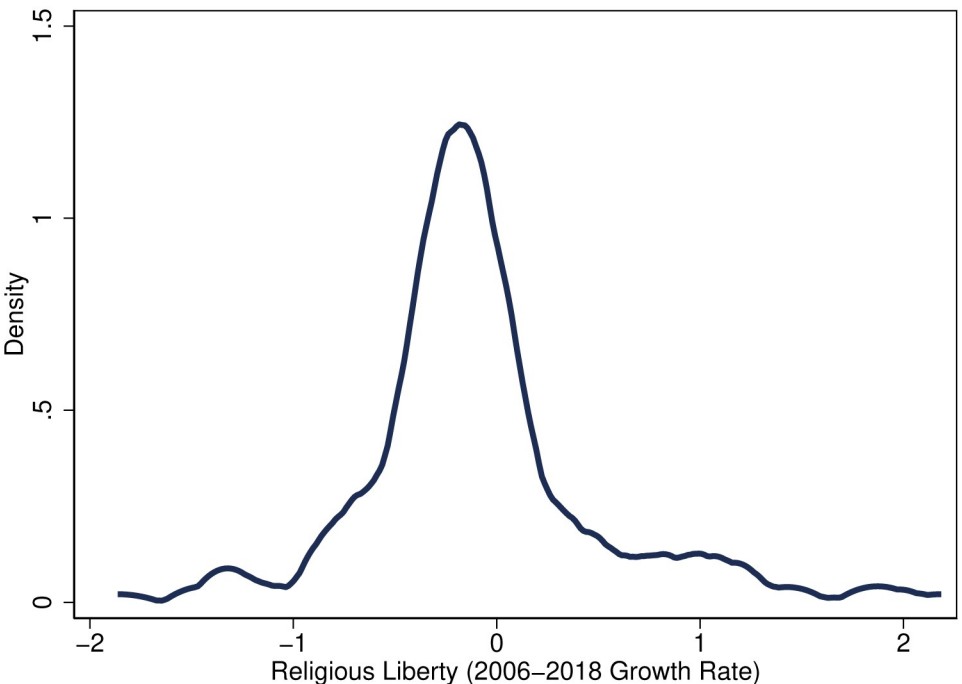

**Fig 4. Distribution in growth in religious liberty, 2006-2018.** Sources: Varieties of Democracy. The figure plots the distribution of growth rates in the religious liberty index between 2006 and 2018 across countries. The sample is trimmed so that there is no observation with a growth rate above 300% or below -300%.

What countries have experienced greater declines in religious liberty over others? Perhaps surprisingly, more developed countries have. Fig 5 investigates the relationship between growth in religious liberty over 2006-2018 and two measures of economic development: the strength of property rights and the logarithm of GDP per capita. While growth in religious liberty exhibits a -0.18 correlation with the strength of property rights, it has a 0.13 correlation with logged GDP per capita. This suggests that, while religious liberty generally expanded in more economically developed countries, it still declined in many countries that tend to rank higher in property rights and the rule of law. For example, Denmark exhibited a 55.6% decline, France a 38.9% decline, the United States a 35.1% decline, and the United Kingdom a 24.9% decline.

## Methodological approach

There is now robust evidence on the cross-sectional correlation between measures of economic development & well-being and religious liberty [3, 4, 46]. While these studies have generally controlled for various country-specific factors, ranging from tax rates to economic activity, such a correlation could still reflect confounding factors that prevent a causal interpretation. This section now outlines a more rigorous approach for causal identification. The main empirical specification, pooling countries observed from 2006 to 2018, examines whether increases in religious liberty are associated with increases in measures of human flourishing through panel regressions of the form:

$$WB_{ict} = \gamma RL_{ct} + \beta D_{it} + \alpha X_{ct} + \eta_c + \lambda_t + \epsilon_{ict} \tag{1}$$

where $WB$ denotes the measures of subjective well-being, $RL$ denotes a standardized measure of religious liberty, $D$ denotes a vector of individual demographic characteristics, $X$ denotes a

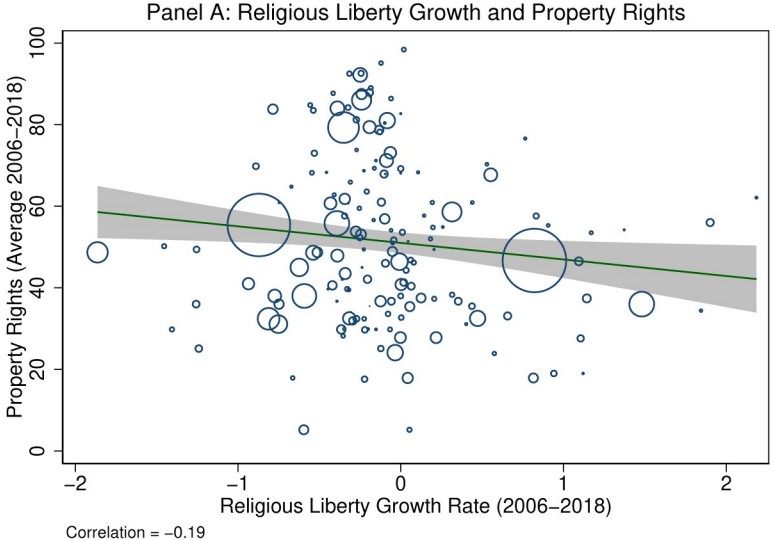

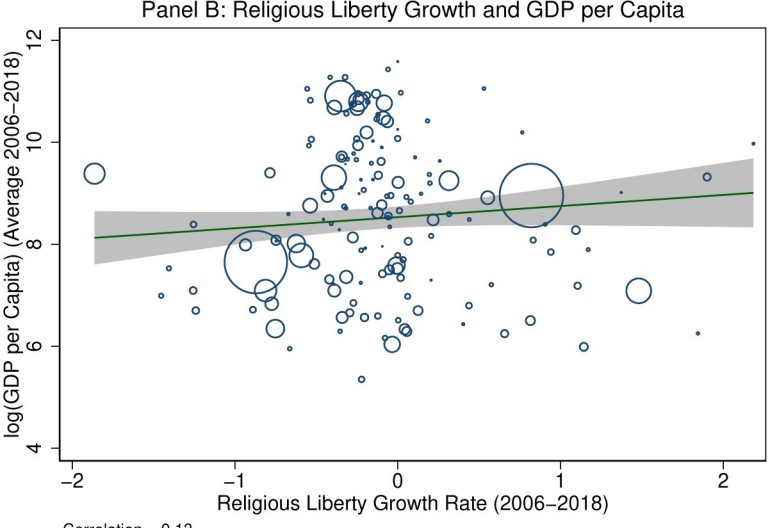

**Fig 5. Growth in religious liberty and measures of economic development.** Sources: Varieties of Democracy, World Bank, and Heritage Foundation. The figure plots the population-weighted relationship between the growth in religious liberty between 2006 and 2018 and an index of property rights and GDP per capita (averaged over 2006 and 2018 levels.

vector of country-level economic and industry factors, and $\eta$ and $\lambda$ denote fixed effects on country and year. To control for measurement error in religious liberty, I also control for the standard deviation of the religious liberty measure. Consistent with the view that measurement error is likely to attenuate the estimates, the results are less statistically significant absent the standard deviation inserted as a control. Standard errors are clustered at the country-level [47].

The inclusion of country and year effects is important for identification since countries vary in remarkably different ways that are often difficult to measure, namely the role of institutions on economic development [6, 10, 18]. In addition to individual demographic characteristics that control for differences in well-being (gender, age, number of children, and education

fixed effects), the inclusion of time-varying country-specific factors, such as GDP growth and industry composition, purges variation in economic activity that could coincide with both political liberalization and human flourishing. For example, if GDP growth increases, then subjective well-being will rise at the same time that political leadership could feel less compelled to regulate religion. Similarly, differences in the agricultural and services employment shares is correlated with a country's stage of economic development. Moreover, to address the possibility that political reforms coincide with economic reforms, indices of property rights, business freedom, and labor freedom from the Heritage Foundation's Index of Economic Freedom are included as controls.

The identifying assumption in Eq 1 is that unobserved shocks to individual well-being are uncorrelated with changes in religious liberty, conditional on various time-varying individual and country characteristics. While prior literature has often taken government and social regulations over religion as exogenous (e.g., [8]), Eq 1 may still produce biased estimates for two reasons. First, to the extent there are other unobserved determinants of well-being that are not captured by the country-specific controls, then estimates of $\gamma$ are upwards biased. Second, given prior research that has found a negative association between measures of religious affiliation and economic growth [8, 9], increases in religious liberty could lead to increases in religious affiliation, producing downwards bias on $\gamma$. These combined forces imply that the naive estimates may be either upwards or downwards biased depending on the strength of one force over another.

Although the country-specific controls are fairly detailed, I also pursue an alternative identification strategy that exploits plausibly exogenous variation in historical exposure to missionaries [11, 48, 49]. In particular, Robert Woodberry constructed a dataset with 143 countries, containing information on the number of Christian missionaries through the 1925 *World Missionary Atlas* [11]. Drawing on his measure of missionary exposure, measured through both Protestant and Catholic missionaries per 10,000 individuals in 1923, I find a strong relationship between them and religious liberty averaged between 2006 and 2018, producing correlations of 0.27 and 0.33, respectively. These results suggest that countries with greater exposure to missionaries (either Protestant or Catholic) prior to 1923 have greater contemporaneous levels of religious liberty.

Importantly, these two variables leverage heterogeneity in the exposure of a country to different types and quantities of missionaries (Protestant and Catholic), which may have had different effects on a country's development of institutions. This is reasonable given that missionaries, especially "conversionary Protestants" (CPs), "had a unique role in spreading mass education, printing, civil society, and other factors that scholars argue fostered democracy" [11]. In particular, rather than building on exploitation from colonial development [6], missionaries publicly advocated for changes in colonial policy, fought for the transfer of ideas, and helped indigenous peoples organize anti-colonial movements [50, 51]. These findings build on a larger literature linking Protestantism and modern representative democracy [52–54].

One reason that could motivate the plausible exogeneity of missionary exposure could stem from the fact that information prior to 1923 diffused much slower than today's news because of the internet. Because conditions across different countries were not common knowledge, it is unlikely that missionaries were optimizing what country to go to in a way that is correlated with future conditions. Moreover, since countries that received more missionaries also required greater support, selection effects could even produce downwards bias since the instrument would predict lower levels of religious liberty due to persistence in negative selection effects. Nonetheless, there is some evidence that missionaries might have gone towards countries that had better conditions, so the IV specifications presented later control for a handful of geographic, climate, and even health and mortality conditions [55].

On the other hand, a separate threat to the identifying assumption is that exposure to missionaries affects contemporaneous well-being through many other channels—not just religious liberty. That is, if exposure leads to more pluralistic and democratic institutions that are also more conducive to growth, better at educating citizens, and/or reduce the propensity for conflict, then the instrument might overestimate the causal effect on religious liberty on well-being. I address this concern by controlling for various factors, such as economic growth and industrial composition, helping isolate the variation that stems from the effect of exposure on religious pluralism.

## Main results and robustness

Table 4 documents the results associated with the baseline specification. Columns 1 and 9 begin by presenting the raw unconditional correlation: a 1sd rise in religious liberty is associated with a 0.04pp increase in the probability that an individual is thriving, but no economically or statistically significant increase in overall well-being. As discussed before, however, the unconditional correlation between religious liberty and human flourishing, particularly for this broader measure, reflects two confounding forces: countries with greater religious liberty also exhibit greater economic freedoms (Fig 3), but greater religious liberty could increase religious affiliation that counteracts economic growth. The subsequent two columns introduce individual demographic and country-specific controls, but the point estimates remain largely indistinguishable from the unconditional correlation reported earlier.

While males report 0.05sd higher levels of well-being, they are 1-2% less likely to report that they are thriving. Individuals who are married are more likely to be thriving have higher well-being, but older individuals and those with families report slightly lower levels. Turning towards education, since the omitted group is those with over a secondary education, the fact that the point estimates on elementary and secondary education are negative suggests that there is a strong association between educational attainment and human flourishing. Turning towards the country-specific controls that begin to appear in columns 3 and 11, there is a positive association between real GDP growth and human flourishing, as is real GDP per capita (not reported) consistent with evidence on the Easterlin hypothesis [56]. Larger countries (at least in population) tend to have lower levels of human flourishing, which could reflect greater competition for scarce resources. Moreover, the agricultural employment share is negatively correlated with human flourishing, which reflects the fact that these countries are at an earlier stage of the development process [57].

Columns 4 and 12 present the baseline specification that contains the standard controls and fixed effects on both country and year, suggesting that 1sd rise in religious liberty is associated with a 0.03pp rise in the probability that an individual is thriving and a 0.08sd rise in individual well-being. The marginal effect of a standard deviation on an indicator for whether an individual is thriving declines to 0.022 (p-value = 0.064) when the sample is restricted to a fully balanced set of countries. Approximately 47% of countries are observed each year from 2006 to 2018, but nearly 81% are observed at least 11 times over those years. Moreover, the number of times that a country is observed is only weakly correlated ($\rho = 0.10$) with the outcome variable, suggesting that measurement error is unlikely to generate bias.

Do these point estimates reflect potential upwards bias emerging from unobserved heterogeneity in the quality of institutions? To investigate the role of unobserved heterogeneity, columns 5-8 and 13-16 sequentially introduce measures of property rights, business freedom, and labor freedom as additional controls. However, these variables tend to produce statistically insignificant estimates, suggesting that they are not culprits behind potential omitted variables bias. Moreover, the fact that the point estimates on religious liberty are not statistically

**Table 4. Baseline results on the effects of religious liberty on human flourishing.**

| | Indicator for Thriving | | | | | | | | | | Overall Subjective Well-being | | | | | |
|---|---|---|---|---|---|---|---|---|---|---|---|---|---|---|---|---|
| | (1) | (2) | (3) | (4) | (5) | (6) | (7) | (8) | (9) | (10) | (11) | (12) | (13) | (14) | (15) | (16) |
| **Main effect:** | | | | | | | | | | | | | | | | |
| Religious liberty | .04*** | .04*** | .02** | .03*** | .03*** | .03*** | .02*** | .03*** | -.00 | .00 | -.00 | .08** | .08** | .09** | .09* | .08** |
| | [.01] | [.01] | [.01] | [.01] | [.01] | [.01] | [.01] | [.01] | [.06] | [.06] | [.04] | [.04] | [.04] | [.04] | [.04] | [.04] |
| **Individual Controls:** | | | | | | | | | | | | | | | | |
| Male | | -.02*** | -.01*** | -.01*** | -.01*** | -.01*** | -.01*** | -.01*** | | .11*** | .11*** | .11*** | .11*** | .11*** | .11*** | .11*** |
| | | [.00] | [.00] | [.00] | [.00] | [.00] | [.00] | [.00] | | [.01] | [.01] | [.01] | [.01] | [.01] | [.01] | [.01] |
| Married | | .02*** | .02** | .01*** | .01*** | .01*** | .01*** | .01*** | | .04* | .03* | .02* | .02* | .02* | .02* | .02* |
| | | [.01] | [.01] | [.00] | [.00] | [.00] | [.00] | [.00] | | [.02] | [.01] | [.01] | [.01] | [.01] | [.01] | [.01] |
| Age | | -.00*** | -.00*** | -.00*** | -.00*** | -.00*** | -.00*** | -.00*** | | -.01*** | -.01*** | -.01*** | -.01*** | -.01*** | -.01*** | -.01*** |
| | | [.00] | [.00] | [.00] | [.00] | [.00] | [.00] | [.00] | | [.00] | [.00] | [.00] | [.00] | [.00] | [.00] | [.00] |
| # of Children | | -.02*** | -.00* | -.00* | -.00*** | -.00*** | -.00*** | -.00*** | | -.04*** | -.03*** | -.03*** | -.03*** | -.03*** | -.03*** | -.03*** |
| | | [.00] | [.00] | [.00] | [.00] | [.00] | [.00] | [.00] | | [.01] | [.01] | [.00] | [.00] | [.00] | [.00] | [.00] |
| Elementary Ed. | | -.25*** | -.17*** | -.15*** | -.15*** | -.15*** | -.15*** | -.15*** | | -.48*** | -.46*** | -.47*** | -.47*** | -.47*** | -.47*** | -.47*** |
| | | [.02] | [.01] | [.01] | [.01] | [.01] | [.01] | [.01] | | [.04] | [.04] | [.02] | [.02] | [.02] | [.02] | [.02] |
| Secondary Ed. | | -.14*** | -.12*** | -.11*** | -.11*** | -.11*** | -.11*** | -.11*** | | -.22*** | -.21*** | -.22*** | -.22*** | -.22*** | -.22*** | -.22*** |
| | | [.01] | [.01] | [.00] | [.00] | [.00] | [.00] | [.00] | | [.02] | [.02] | [.02] | [.02] | [.02] | [.02] | [.02] |
| **Country Controls:** | | | | | | | | | | | | | | | | |
| Real GDP Growth | | | .13 | .09 | .08 | .08 | .08 | .09 | | | 2.61*** | 1.01*** | .97*** | .93*** | .95*** | .96*** |
| | | | [.12] | [.06] | [.07] | [.07] | [.06] | [.06] | | | [.55] | [.29] | [.30] | [.29] | [.29] | [.29] |
| log(Population) | | | .02*** | -.02 | -.01 | -.02 | -.01 | -.01 | | | .06*** | -1.10*** | -1.00*** | -1.04*** | -1.03*** | -1.00*** |
| | | | [.00] | [.07] | [.07] | [.07] | [.07] | [.07] | | | [.02] | [.30] | [.30] | [.30] | [.30] | [.31] |
| Agriculture Emp., % | | | .01 | -.91*** | -.92*** | -.91*** | -.86*** | -.85*** | | | .68* | -1.23 | -1.15 | -1.22 | -1.12 | -1.11 |
| | | | [.13] | [.23] | [.24] | [.23] | [.24] | [.24] | | | [.41] | [.87] | [.86] | [.86] | [.87] | [.87] |
| Service Emp., % | | | .63*** | -1.06*** | -.99*** | -.99*** | -.95*** | -.94*** | | | 1.37** | -1.45 | -1.48 | -1.46 | -1.39 | -1.44 |
| | | | [.18] | [.29] | [.29] | [.29] | [.30] | [.30] | | | [.53] | [1.24] | [1.24] | [1.24] | [1.25] | [1.24] |
| Property Rights | | | | | .00 | | | -.00 | | | | | .03 | | | .03 |
| | | | | | [.01] | | | [.01] | | | | | [.03] | | | [.03] |
| Business Freedom | | | | | | .01 | | .01 | | | | | | -.01 | | -.01 |
| | | | | | | [.01] | | [.01] | | | | | | [.03] | | [.03] |
| Labor Freedom | | | | | | | .01** | .01** | | | | | | | .02 | .02 |
| | | | | | | | [.01] | [.01] | | | | | | | [.02] | [.02] |
| R-squared | .03 | .08 | .12 | .17 | .17 | .17 | .17 | .17 | .00 | .04 | .05 | .10 | .10 | .10 | .10 | .10 |
| Sample Size | 1593390 | 1500094 | 1471633 | 1471633 | 1451924 | 1458659 | 1458659 | 1451924 | 1593390 | 1500094 | 1471633 | 1471633 | 1451924 | 1458659 | 1458659 | 1451924 |
| Individual Controls | No | Yes | Yes | Yes | Yes | Yes | Yes | Yes | No | Yes | Yes | Yes | Yes | Yes | Yes | Yes |
| Country Controls | No | No | Yes | Yes | Yes | Yes | Yes | Yes | No | No | Yes | Yes | Yes | Yes | Yes | Yes |
| Country FE | No | No | No | Yes | Yes | No | Yes | Yes | No | No | No | Yes | Yes | Yes | Yes | Yes |

*(Continued)*

**Table 4.** (Continued)

| | Indicator for Thriving | | | | | | | | Overall Subjective Well-being | | | | | | | |
|---|---|---|---|---|---|---|---|---|---|---|---|---|---|---|---|---|
| | (1) | (2) | (3) | (4) | (5) | (6) | (7) | (8) | (9) | (10) | (11) | (12) | (13) | (14) | (15) | (16) |
| Year FE | No | No | No | Yes | Yes | No | Yes | Yes | No | No | No | Yes | Yes | Yes | Yes | Yes |

Sources: Gallup World Poll, Heritage Foundation, World Bank, Varieties of Democracy, 2006-2018. The table reports the coefficients associated with regressions of human flourishing on a standardized z-score of religious liberty, conditional on controls and country and year fixed effects. The first measure is an indicator for whether an individual is thriving, which is generated based on having a response of 7 or higher on a 10-point scale about current life satisfaction and having a response of 8 or higher on a similar 10-point scale about expected future life satisfaction in 5 years; the second measure is constructed using a principal component analysis (PCA) on four standard normal measures of subjective well-being: daily experience, optimism, positive experience, and negative experience (see Table 1). Individual controls include: an indicator for male, marital status, age, number of children, education (elementary and secondary, normalized to over secondary as the omitted group). Country controls include: real GDP growth, logged population, agricultural employment share, service employment share, and standardized z-scores of property rights, business freedom, and labor market freedom. Observations are weighted by the appropriate sample weights and all standard errors are clustered at the country-level.

**Table 5. Examining the effects of religious liberty on economic development.**

|  | log(Real GDP) | | GDP growth | | School enrollment | |
|---|---|---|---|---|---|---|
|  | (1) | (2) | (3) | (4) | (5) | (6) |
| Religious liberty (z-score) | .035 | .020 | -.005*** | .010 | .025 | .022* |
|  | [.050] | [.022] | [.002] | [.007] | [.015] | [.013] |
| Economic freedom (z-score) | .311*** | .096*** | .001 | -.008*** | .014 | -.020* |
|  | [.055] | [.020] | [.002] | [.003] | [.016] | [.010] |
| log(Population) | .998*** | .467*** | -.000 | -.023 | .001 | .060 |
|  | [.033] | [.089] | [.001] | [.018] | [.007] | [.056] |
| Agricultural share | -3.582*** | -2.227*** | -.029 | -.097*** | -1.019*** | -.721*** |
|  | [.739] | [.495] | [.018] | [.029] | [.149] | [.195] |
| Services share | 1.761* | -1.310** | -.081*** | -.152*** | -.026 | -.308* |
|  | [1.023] | [.613] | [.023] | [.048] | [.212] | [.168] |
| R-squared | .90 | 1.00 | .05 | .23 | .70 | .96 |
| Sample Size | 3543 | 3542 | 3539 | 3539 | 2489 | 2483 |
| Country Controls | Yes | Yes | Yes | Yes | Yes | Yes |
| County FE | No | Yes | No | Yes | No | Yes |
| Year FE | No | Yes | No | Yes | No | Yes |

Sources: World Bank, Heritage Foundation, Varieties of Democracy, 1995-2018. The table reports the coefficients associated with regressions of logged real GDP, GDP growth, and school enrollment in secondary education as a percent of gross enrollment on standardized religious liberty, controlling for various time-varying country characteristics. Standard errors are clustered at the country-level and observations are unweighted.

different from the baseline suggests that any unobserved determinants of human flourishing are likely only weakly correlated with religious liberty [58].

Are these gains in individual well-being the result of the indirect effect of the potentially beneficial effects of religious liberty on economic development? While various time-varying controls were included in the earlier results, I now investigate regressions of economic activity on religious liberty under similar specifications. Table 5 documents these results. Both the OLS and FE estimates are presented for completeness, but the FE estimates are preferred because they remove potentially problematic time-varying unobserved heterogeneity.

There is a positive, but statistically insignificant at conventional levels, effect of improvements in religious liberty on real GDP and GDP growth. This could be a result of endogeneity emerging from the potentially negative contemporaneous effect of religious affiliation on productivity, but it nonetheless suggests that the results from Table 4 are unlikely plagued by these concerns about omitted variables. There is also a positive, and statistically significant at the 10% level, effect of religious liberty on school enrollment in secondary education as a share of gross enrollment. This is consistent with the view that increased religious pluralism encourages greater pursuit of educational attainment and creativity, but no causal interpretation can be ascribed to these results.

To understand whether these results reflect a genuine causal effect between religious liberty and human flourishing, a useful placebo test involves examining whether the improvements in human flourishing are concentrated among religious individuals. Using information on religious affiliation, which is available for roughly 85% of the sample, I estimate a modified version of Eq 1 that allows for an interaction between a measure of religious affiliation and religious liberty:

$$WB_{ict} = \gamma RL_{ct} + \phi r_{ict} + \xi(r_{ict} \times RL_{ct}) + \beta D_{it} + \alpha X_{ct} + \eta_c + \lambda_t + \epsilon_{ict} \qquad (2)$$

where now $r$ denotes an indicator for religious affiliation. I focus on two measures: whether the individual considers themselves religious and whether the individual considers themselves a Christian. Letting the outcome variable denote overall well-being, I find that $\hat{\xi} = 0.028$ ($p$-value = 0.225) for individuals who identify as religious and $\hat{\xi} = 0.124$ ($p$-value = 0.00) for individuals who identify as Christian or Jewish. This suggests that, while religious individuals benefit overall from improvements in religious liberty, the benefits are concentrated among Christians and Jews. One potential explanation behind this result arises from the fact that Christians and Jews are persecuted in many countries that rank low on levels of religious liberty, such as the Middle East and China, meaning that they are likely the largest beneficiaries of improvements in religious liberty.

However, these fixed effects specifications could still produce downwards biased estimates of the causal effect of religious liberty on human flourishing because of the negative association between religious affiliation and economic growth [8, 9]. To address these concerns, I now exploit plausibly exogenous historical variation in exposure to missionaries prior to 1923. The identifying assumption is that, after controlling for contemporaneous country-specific factors, such as economic activity and institutional quality, the effect of missionaries on human flourishing operates only through its effects on improvements in religious liberty.

The exclusion restriction could be violated in several ways. If missionary exposure affects economic development through other channels, then we might overestimate the effects of missions on human flourishing. To address the possibility, I control for geographic characteristics (whether the country is an island, whether it is landlocked, and latitude), average temperature in the hottest and coldest month, and whether there was a malaria epidemic in the country, on top of the standard individual and contemporaneous country controls that are in the baseline specification [11]. These controls also address the possibility that certain countries might have had better transportation networks or a lower incidence of disease, which could affect the attractiveness or ability for missionaries to go to the country in the first place [55]. Subsequent work has explored answers to these concerns about omitted variables in much greater detail [59].

Turning towards these IV results, I find that a 1sd rise in religious liberty is associated with a 0.084 percentage point ($p$-value = 0.002) rise in the probability that an individual reports that they are thriving, which is larger than the marginal effect of 0.018 in the baseline specification. While the $F$-statistic is only 7, thus below the rule-of-thumb of 10, the marginal effect is nonetheless statistically significant. However, because the instrument is cross-sectional, standard errors are now clustered at the country-by-year level. Moreover, if additional controls on mortality status, life expectancy, urbanization, and population density as of 1500 are added as additional historical controls, the marginal effect declines to 0.023, which is statistically indistinguishable from the 0.018 in the earlier results. In this sense, the IV results provide complementary support that the baseline specification reflects a genuine causal effect.

## Understanding the mechanisms

There are at least three mechanisms that could explain the observed positive effects. First, it is possible that religious freedom creates the seeds for democracy by producing a space where self-expression and public discourse is free and open. Empirical evidence suggests that these factors were important for the emergence and continuity of democracy in Europe and North America [52–54]. Although some argue that democracy was developed purely through Enlightenment ideas, countries that developed democracies purely on the basis of Enlightenment ideals were generally not stable [60] and/or consisted of only the elites [61, 62]. For example, there is evidence, particularly among conversionary Protestants, that religious

movements and the presence of religious freedom led to the development of newspapers and print media through the printing press, which leveraged public opinion to democratize society and decentralize power from the elites [63].

Second, another way that religious pluralism could be linked with improvements in well-being is through its effects on educational attainment. For example, much like elites in the nineteenth century resisted educating women and the poor based on the concern that it would lead to instability [64, 65], the promotion of religious liberty is driven by the belief that each individual has the option and responsibility of pursuing truth and becoming educated on their terms, rather than by force [66]. The resulting improvements in educational attainment could have driven improvements in per capita income growth and human flourishing [67].

Third, religious freedom could mitigate the incidence of conflict and civil strife by promoting greater respect for one another and processes for resolving disputes. For example, many of the non-violent tactics for social reform, such as boycotts and mass petitions, were piloted by religious organizations [68, 69]. Conversionary Protestants were especially common organizers behind movements throughout the world, ranging from Great Britain [68, 70] to the United States [51] to India [71, 72]. [50]. At the core of these movements was the belief that all individuals are created equal with the ability and interest to thrive if given the opportunity, countering many of the colonial and class-specific narratives that prevailed. Protestant Christians were particularly likely to enter and positively influence civil society by enacting reforms and promoting peace [73–76]. Moreover, by promoting equality across traditional race and class structures, religious forces were more likely to create and maintain stable democracies and survive potential authoritarian regimes that take over [77].

To investigate the relative importance of these competing mechanisms, Table 6 considers both standard least squares and fixed effects estimates to capture cross-sectional and within-country differences. Moreover, because these exercises make greater use of the World Bank and V-Dem data, the sample can extend further back to 1995 up to 2018. The sample is fairly balanced with 165 to 168 countries each year. Starting with measures of educational attainment, there is positive association between religious liberty and schooling in the cross-section: a 1sd rise in religious liberty is associated with a 0.03pp and 0.06pp rise in secondary and tertiary school enrollment, but no association with primary school enrollment. Given that the averages are 0.77 and 0.35, respectively, the estimate for tertiary school is economically meaningful: a marginal effect amounts to 17% (= 0.06/.35) of the mean. However, once country and year fixed effects are introduced, these correlations vanish. One interpretation is that the cross-section captures a long-term effect that religious liberty may have on well-being through changes in the education system, whereas the year-to-year variation is capturing more of the short to medium -run impact.

Turning towards measures of corruption as a proxy for democratic governance, there is a strong moderating effect: a 1sd rise in religious liberty is associated with a 0.29sd and 0.32sd decline in political and public corruption, respectively, in the cross-section and a 0.24sd and 0.18sd decline within a country over time. The fact that the estimate is stronger for political corruption is consistent with theory and evidence about the link between civic participation and democratic governance [78]: religious liberty and freedom of thought empowers individuals and encourages greater civic participation, thereby leading to greater accountability and transparency.

Turning towards more direct measures of democratic governance based on the exercise of freedom, a 1sd rise in religious liberty is associated with a 0.59sd, 0.30sd, 0.34sd, and 0.55sd rise in civil liberties, women empowerment, access to justice, and freedom of expression, respectively, using only the within-country variation. The cross-sectional estimates are roughly 25-80% larger than the fixed effects estimates. The fact that improvements in religious liberty are

**Table 6. Examining the mechanisms behind the effects of religious liberty on well-being.**

| | Primary School | | Secondary School | | Tertiary School | | Political Corruption | | Public Corruption | |
|---|---|---|---|---|---|---|---|---|---|---|
| | (1) | (2) | (3) | (4) | (5) | (6) | (7) | (8) | (9) | (10) |
| *Panel A* | | | | | | | | | | |
| Freedom of religion | .02 | -.00 | .03** | .02 | .06*** | -.02 | -.29*** | -.24** | -.32*** | -.18* |
| | [.01] | [.02] | [.01] | [.01] | [.01] | [.02] | [.05] | [.10] | [.05] | [.10] |
| R-squared | .04 | .73 | .71 | .96 | .59 | .93 | .49 | .96 | .52 | .95 |
| Sample Size | 3098 | 3097 | 2600 | 2597 | 2439 | 2436 | 3825 | 3825 | 3835 | 3835 |
| | Civil Liberties | | Women Empowerment | | Access to Justice | | Freedom of Expression | | Armed Conflict | |
| *Panel B* | | | | | | | | | | |
| Freedom of religion | .72*** | .59*** | .65*** | .30*** | .53*** | .34*** | .67*** | .55*** | -.00 | -.05 |
| | [.05] | [.07] | [.07] | [.05] | [.07] | [.06] | [.06] | [.09] | [.02] | [.08] |
| R-squared | .68 | .94 | .56 | .94 | .49 | .95 | .57 | .91 | .08 | .77 |
| Sample Size | 3835 | 3835 | 3804 | 3804 | 3835 | 3835 | 3835 | 3835 | 920 | 919 |
| Country Controls | Yes | Yes | Yes | Yes | Yes | Yes | Yes | Yes | Yes | Yes |
| Country FE | No | Yes | No | Yes | No | Yes | No | Yes | No | Yes |
| Year FE | No | Yes | No | Yes | No | Yes | No | Yes | No | Yes |

Sources: Varieties of Democracy, 1990-2018. The table reports the coefficients associated with regressions of various dimensions of country performance on a standardized *z*-score of religious liberty, conditional on country controls, including: GDP growth, logged population, the agricultural employment share, the services employment share, and the standard deviation on the estimate of religious liberty from V-Dem. Outcome variables include: primary school enrollment as a share of gross enrollment, secondary school enrollment as a share of gross enrollment, tertiary school enrollment as a share of gross enrollment, standardized *z*-scores on political corruption, public corruption, civil liberties, women empowerment (specifically in politics), access to justice, freedom of expression, and an indicator of armed conflict (either external or internal). Standard errors are clustered at the country-level.

so economically and statistically associated with improvements in these dimensions of freedom and civil liberties suggests it is the primary mechanism linking religious liberty and well-being. Finally, while there is a negative association between the probability of being in an armed conflict and improvements in religious liberty, the association is statistically insignificant.

## Conclusion and policy implications

While these exercises do not provide a silver-bullet explanation behind the plausibly causal effect of improvements in religious liberty on dimensions of human flourishing, they suggest that religious liberty—and, more broadly, social capital [79]—and democratic institutions are complements, leading to greater freedom of expression and civil liberties for a country's people. Absent the basic human right for individuals to believe and worship freely, it is hard to imagine how a country can promote economic and social prosperity: suppression of thought will necessarily inhibit entrepreneurship, innovation, and social welfare more broadly.

Using the most comprehensive database to date on measures of human flourishing across time and space, this paper quantifies the effects of religious liberty, finding that a 1sd rise in religious liberty is associated with a 0.03pp rise in the probability an individual is thriving and a 0.08sd rise in overall well-being. The baseline identification strategy exploits plausibly exogenous year-to-year changes in religious liberty restrictions, controlling for a wide array of demographic and country-specific factors, such as GDP growth and institutional quality. To

understand their robustness, an alternative instrumental variables strategy that exploits historical variation in the exposure of countries to missionaries prior to 1923 suggests slightly larger estimates on human flourishing.

These results show that religious liberty should also be taken into account when constructing and evaluating measures of economic development and welfare. Indeed, Alexis de Tocqueville in *Democracy in America* argued that religious liberty in America would be fundamental to its democratic governance and preservation of peace and stability, balancing the competing demands for materialism and religious fanaticism.

The results are particularly timely given the competition of values between the United States and the Chinese Communist Party (CCP) where, for example, religious minorities in China are persecuted [80]. Moreover, given the unfolding COVID-19 pandemic and the international backlash against their failure to warn against the spread of the virus [81, 82], the CCP has a unique opportunity to not only advance its own economic development through an expansion of religious liberty, but also signal to the international community that it is willing to make reasonable concessions on important human rights issues [46].

## Acknowledgments

Thank you to Brian Grim, Byron Johnson, Gale Pooley, and Tyler VanderWeele for comments and suggestions. All errors and views are my own and do not represent any affiliated institutions.

## Author Contributions

**Conceptualization:** Christos Andreas Makridis.

**Data curation:** Christos Andreas Makridis.

**Formal analysis:** Christos Andreas Makridis.

**Funding acquisition:** Christos Andreas Makridis.

**Investigation:** Christos Andreas Makridis.

**Methodology:** Christos Andreas Makridis.

**Project administration:** Christos Andreas Makridis.

**Resources:** Christos Andreas Makridis.

**Software:** Christos Andreas Makridis.

**Supervision:** Christos Andreas Makridis.

**Validation:** Christos Andreas Makridis.

**Visualization:** Christos Andreas Makridis.

**Writing – original draft:** Christos Andreas Makridis.

**Writing – review & editing:** Christos Andreas Makridis.

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
