## [Decision Letter · Decision Letter 0]

30 Jun 2020

PONE-D-20-10710

Human Flourishing and Religious Liberty: Evidence from Over 150 Countries, 2006-2018

PLOS ONE

Dear Dr. Makridis,

Thank you for submitting your manuscript to PLOS ONE. After careful consideration, we feel that it has merit but does not fully meet PLOS ONE’s publication criteria as it currently stands. Therefore, we invite you to submit a revised version of the manuscript that addresses the points raised during the review process.

Please see my additional comments to the author attached below.

We look forward to receiving your revised manuscript.

Kind regards,

Yanyan Gao

Academic Editor

PLOS ONE

Journal Requirements:

'The funders had no role in study design, data collection and analysis, decision to publish, or preparation of the manuscript.'

Additional Editor Comments (if provided):

Two reviewers, especially reviewer no.2, have provided detailed comments on the manuscript. They both acknowledge the merits of the study that support publication, while also propose some important issues to be addressed. Thus, I encourage the author to revise and resubmit the paper. The acceptance of the paper will depend on the extent that these comments can be addressed or effectively defended as well as the results of the second-round review.

Reviewers' comments:

Reviewer's Responses to Questions

**Comments to the Author**

1. Is the manuscript technically sound, and do the data support the conclusions?

Reviewer #1: Yes

Reviewer #2: Yes

2. Has the statistical analysis been performed appropriately and rigorously? 

Reviewer #1: Yes

Reviewer #2: Yes

3. Have the authors made all data underlying the findings in their manuscript fully available?

Reviewer #1: Yes

Reviewer #2: No

4. Is the manuscript presented in an intelligible fashion and written in standard English?

Reviewer #1: Yes

Reviewer #2: Yes

5. Review Comments to the Author

Reviewer #1: Your introduction is strong and establishes the importance and timeliness of your topic and analysis. The statistical analysis is rigorous and appropriate to the questions being investigated. The data are from available, well-known and respected data sources. The manuscript is well written, clear and accessible to specialists and non-specialists alike.

Specific comments:

1. Grim, Clark and Snyder's 2014 study actually went beyond correlational relationship between religious freedom and economic growth, empirically testing and finding the tandem effects of religious restrictions and hostilities to be detrimental to economic growth while controlling for other theoretical, economic, political, social, and demographic factors. This should be specifically mentioned, especially since your findings are in line with those and expand them beyond mere GDP growth to human flourishing. (see attached file)

2. In your final section, it would be very helpful to split out the "conclusions" from "understanding the mechanisms" by having as the conclusion a bulleted summary of your main findings and their implications for societies.

3. You mention the importance of China being a respected democratic partner as your very last point. However, since your whole paper is about flourishing, I strongly suggest your comment on China should focus on pointing out that if the Chinese Communist Party (i.e., not 'China' - the two are not synonymous) were to make religious freedom a priority and a reality for all, it would be a significant asset to China's economic recovery and the flourishing of its people following the pandemic. Moreover, such a move would also help address the growing global backlash toward China following the pandemic.

Regarding the backlash, see: "Coronavirus: how will China’s role in the global economy change when faced with pandemic backlash?" Zhou Xin, 28 Apr, 2020, SCMP, https://www.scmp.com/economy/china-economy/article/3081778/coronavirus-how-will-chinas-role-global-economy-change-when

Reviewer #2: This is an interesting paper which assesses the spatial and temporal changes in religious liberty and attempts to address the difficult question of the effect of religious liberty on measures of human flourishing. The study uses standard panel data techniques and an instrumental variable approach to address the identified research questions. The writing for the most part is concise and clear, and the literature that is cited is relevant. However, the article in its current form needs some improvements if its arguments and policy implications are to carry credence. I divide my concerns into major and minor themes.

I suggest four major themes for improvement. First, the causal claims that this paper makes need to be argued better. The introduction on page 4 introduces the reader to the three possible mechanisms by which religious liberty could impact well-being. However, their expansion in Section 5 ‘Understanding the mechanism and conclusion’ (pp.20-21) is not convincing. The first example of the link between religious factors and the emergence of democracy, does not show how religious plurality or religious freedom contributes to democracy. Instead it presents evidence of a specific religious movement contributing towards democracy. The second example connecting religious plurality and educational attainment is relatively more convincing. In case of the third causal mechanism, again the examples cited reflect the fact that specific religious movements contributed to the promotion of peace and positively influenced civil society. There is a body of literature that points to the link between religion and pro-social behavior, however, the focus of this study is not religiosity rather it is religious pluralism. Thus, there is a strong need for the causal claims to be supported and positioned better.

Second, the level of detail the paper provides with respect to the data and methodology and the organisation of this information needs to be improved upon considerably. This begins from the abstract itself, where it is not clear how many countries feature in the first part of the analysis and which is the time-period under consideration. Is it the same set of countries that are used in the analysis that are then used in the second part of the analysis that uses micro-data?

The introduction gives the reader some semblance of the time-period of analysis (2006-2018). But it is still unclear how many countries are part of this analysis and which dataset is used in the first part of the analysis and which in the second. Providing this information upfront in the introduction itself is important, as the paper states (on page 6) that one of its contributions is that unlike past studies which use the World Values Data, this particular study employs a much more comprehensive dataset.

On page 11, Table 3 mentions “agricultural share” and “services share”, but it is unclear whether this implies the sectoral share with respect to the GDP or with respect to the labor force. It is only much later on page 17, where it is specified that agricultural share means agricultural employment share.

On page 14, the paper states: “Moreover, to address the possibility that political reforms coincide with economic reforms, proxies for economic freedom from the Heritage Foundation’s Index of Economic Freedom are included as controls.” It would be good to mention what these proxies are.

Another example, on page 15, the study states: ““Drawing on his measure of missionary exposure, measured through both Protestant and Catholic missionaries per 10,000 individuals in 1923, Figure 6 plots their relationship with religious liberty averaged between 2006 and 2018, producing correlations of 0.27 and 0.33, respectively.” This statement does not shed light on essential information such as the source of this data, nor does it tell the reader the geographic coverage or time-period of reporting. I had to read through the footnote of Table 6 (on page 41) to find the answer to my question. It would be more effective if this information were to be reported in the main text itself.

Finally, Table 5 is supposed to report the OLS and fixed effects estimates. It is not clear which column refers to which. A little more attention to detail would have been good.

Third, this point relates to the policy implications that the study draws. According to the paper the causal mechanism by which increases in religious liberty lead to increases in well-being is through the links between religious liberty (social capital) and democracy. But then this very causal claim would belie the policy implication that the study cites by giving the example of China. If religious liberty is related to increases in democratic governance, would authoritarian countries like China be interested in pursuing it? Moreover, what would explain the case of a country like Singapore where religious liberty co-exists with a lack of an actual democracy? But while the latter question may be a topic for future research, the current study might need to give some thought to the policy implications it draws. Perhaps it would be more convincing to draw policy implications around the premise that if religious liberty helps increase human flourishing, then the recent decline of religious liberty in countries that rank higher in property rights and the rule of law, does not bode well.

Fourth, this point is with respect to the analysis. I have some questions related to the controls, the instrumental variable and the spatial-temporal coverage of the analysis. Some minimum rationale for the controls needs to be provided. For instance, what is the justification for including specifically the share of agriculture and services as controls, in the main regressions. Why not industry and services?

Some recent papers have also questioned the supposed exogeneity of missionary activity and its use an instrumental variable. Jedwab et al. (2018) argue that missionary’s locational activity decisions were driven by economic factors as missionaries went to healthier, safer and more accessible and developed areas first. Thus, locations with past missions tended to be more developed and this bias is further enhanced as historical mission atlases tend to only report the best mission locations. The same study also quotes a missionary Thauren in Togo who describes the settlement strategy as one where every effort is made to acquire knowledge of the region (p.2). Evidence such as this lowers the plausibility of the instrument, but if perhaps the article were to support its instrumental variable strategy with some basic tests for IVs such as by quoting the Stock-Watson statistics etc., it would make the results more believable.

Finally, how many of the countries from the original analysis feature in the instrumental variable analysis? Also, the first two parts of the analysis use a dataset between 2006-2018, but in tracing the causal mechanisms the time-period of analysis is from 1990-2018. Is there a particular reason why a different time-period was chosen for this part of the analysis?

I now list below my minor comments.

1. Page 6: “98% of the world’s population”, the author probably means 150 countries that make up 98% of the world’s population. But worded the way it is at present, it seems as if the World Gallup Poll surveys 98% of the world population.

2. Pages 6-7: What is the average sample size that Gallup uses for a country? It would be useful to present this information.

3. It is mentioned that the V-Dem dataset measures responses on a 0-4 scale and then rescales them. It would be good to mention the new scale in the text.

4. Page 8: “In this sense, while the Gallup data provides several comparable measures to the WVS, it also provides a wide range of additional responses about institutions, human ﬂourishing, and religious aﬃliation.” In what sense does it provide information on institutions?

5. Page 8-9: Pew research centre data is mentioned here. Its mention here necessitates an explanation for what it does that makes it inferior to the V-dem dataset.

6. There is some repetition on pages 9 and 10.

Page 9: “Moreover, because experts provide ratings across more than one categories of questions, those who systematically deviate from the norm will receive a lower reliability rating, which leads to a lower weight in the construction of the overall index.”

Page 10: “Because experts in the V-Dem survey answer multiple questions, multiple

answers that deviate from the norm will produce a lower reliability, thereby generating lower weights in the inclusion of the respondent’s answer in the overall score.”

7. Page 11: How does pew research centre define restrictions and social hostilities?

8. Page 11: “There are many approaches for measuring economic freedom, most notably the Index of Economic Freedom [46]”. Does this refer to the Heritage Foundation Index of Economic Freedom?

9. Page 17: “Moreover, the agricultural employment share is negatively correlated with human ﬂourishing, which reﬂects the fact that lower income countries have greater agricultural employment shares [57]”. This sentence could be worded to make it more clear. Is the point that countries with higher agricultural employment shares are those where human flourishing is reported to be lower, these also tend to be low income countries?

10. Pages 17 and 23, use the term probability in interpreting the results. For example on page 17: ““Columns 4 and 12 present the baseline speciﬁcation that contains the standard controls and ﬁxed eﬀects on both country and year, suggesting that 1sd rise in religious liberty is associated with a 0.03pp rise in the probability that an individual is thriving and a 0.08sd rise in individual well-being”. Is that correct?

11. Page 19: “This suggests that, while religious individuals beneﬁt overall from improvements in religious liberty, the beneﬁts are concentrated among Christians.” This is confusing because the preceding and subsequent sentence mention Christians or Jews together. And then this sentence restricts its scope to Christians.

12. Pages 34-35: Tables 1 and 2 have a problem with the labeling.

13. Page 34: Table 1’s footnote seems incomplete. It does not cover the case of questions where there are 11 items, for example the “Daily Experience Index”.

14. Page 35: It would be helpful if for Table 2, the table/ appended note mentioned the precise question that was asked.

15. Pages 33-42: There are issues with the order of the tables and figures. If they are supposed to appear in the order they are mentioned in the text, then table 3 is out of place.

16. Where does the data for the estimations in Section 5 come from i.e. the data on armed conflict etc.? Is it from V-dem?

Reference

Jedwab, Remi, Meier zu Selhausen, Felix and Moradi, Alexander (2018) The economics of missionary expansion: evidence from Africa and implications for development. Working Paper. University of Oxford, Oxon.

6. PLOS authors have the option to publish the peer review history of their article (what does this mean?). If published, this will include your full peer review and any attached files.

Reviewer #1: Yes: Brian J Grim, Ph.D.

Reviewer #2: No

---

## [Decision Letter · Decision Letter 1]

15 Sep 2020

PONE-D-20-10710R1

Human flourishing and religious liberty: Evidence from over 150 countries

PLOS ONE

Dear Dr. Makridis,

Thank you for submitting your manuscript to PLOS ONE. After careful consideration, we feel that it has merit but does not fully meet PLOS ONE’s publication criteria as it currently stands. Therefore, we invite you to submit a revised version of the manuscript that addresses the points raised during the review process.

As you can see below, both reviewers recommended to accept your manuscript for publication. However, the second reviewer proposed a new comment which, I also think, should be further addressed. Please make sure that all your implications are developed with an unbiased, arguable position and are closely related to your findings. After this, I will proceed the manuscript to be fully accepted for publication.

We look forward to receiving your revised manuscript.

Kind regards,

Yanyan Gao

Academic Editor

PLOS ONE

Reviewers' comments:

Reviewer's Responses to Questions

**Comments to the Author**

1. If the authors have adequately addressed your comments raised in a previous round of review and you feel that this manuscript is now acceptable for publication, you may indicate that here to bypass the “Comments to the Author” section, enter your conflict of interest statement in the “Confidential to Editor” section, and submit your "Accept" recommendation.

Reviewer #1: All comments have been addressed

Reviewer #2: All comments have been addressed

2. Is the manuscript technically sound, and do the data support the conclusions?

Reviewer #1: Yes

Reviewer #2: Yes

3. Has the statistical analysis been performed appropriately and rigorously? 

Reviewer #1: Yes

Reviewer #2: Yes

4. Have the authors made all data underlying the findings in their manuscript fully available?

Reviewer #1: Yes

Reviewer #2: Yes

5. Is the manuscript presented in an intelligible fashion and written in standard English?

Reviewer #1: Yes

Reviewer #2: Yes

6. Review Comments to the Author

Reviewer #1: Important findings with clear economic and policy implications. This confirms that religious freedom is actually increasing as an asset to stability, wellbeing and prosperity.

Reviewer #2: Thank you for your detailed responses to each of the comments. These were very helpful. With the revisions the paper looks good to go. I only have one last suggestion and that would be to cite a source for the comment you make in your concluding paragraph regarding the backlash against China with respect to the COVID-19 pandemic.

Thank you for this very interesting and well-written article. I look forward to seeing the published version!

7. PLOS authors have the option to publish the peer review history of their article (what does this mean?). If published, this will include your full peer review and any attached files.

Reviewer #1: **Yes: **Brian J Grim

Reviewer #2: No

---

## [Editor Report · Decision Letter 2]

17 Sep 2020

Human flourishing and religious liberty: Evidence from over 150 countries

PONE-D-20-10710R2

Dear Dr. Makridis,

We’re pleased to inform you that your manuscript has been judged scientifically suitable for publication and will be formally accepted for publication once it meets all outstanding technical requirements.

Kind regards,

Yanyan Gao

Academic Editor

PLOS ONE
---

## [Editor Report · Acceptance letter]

22 Sep 2020

PONE-D-20-10710R2 

Human flourishing and religious liberty: Evidence from over 150 countries 

Dear Dr. Makridis:

I'm pleased to inform you that your manuscript has been deemed suitable for publication in PLOS ONE. Congratulations! Your manuscript is now with our production department. 

Kind regards, 

on behalf of

Dr. Yanyan Gao 

Academic Editor

PLOS ONE